# Neurofilament Light Chains in Systemic Amyloidosis: A Systematic Review

**DOI:** 10.3390/ijms25073770

**Published:** 2024-03-28

**Authors:** Milou Berends, Hans L. A. Nienhuis, David Adams, Chafic Karam, Marco Luigetti, Michael Polydefkis, Mary M. Reilly, Yoshiki Sekijima, Bouke P. C. Hazenberg

**Affiliations:** 1Department of Internal Medicine, Amyloidosis Center of Expertise, University Medical Center Groningen, 9700 RB Groningen, The Netherlands; m.berends@umcg.nl (M.B.); h.l.a.nienhuis@umcg.nl (H.L.A.N.); 2Service de Neurologie, CHU Bicêtre, Assistance Publique—Hôpitaux de Paris, University Paris-Saclay, CERAMIC, Le Kremlin-Bicêtre, 94270 Paris, France; david.adams@aphp.fr; 3Department of Neurology, University of Pennsylvania, Philadelphia, PA 19104, USA; chafic.karam@pennmedicine.upenn.edu; 4UOC Neurologia, Fondazione Policlinico A. Gemelli IRCCS, 00168 Rome, Italy; mluigetti@gmail.com; 5Dipartimento di Neuroscienze, Università Cattolica del Sacro Cuore, 00168 Rome, Italy; 6Department of Neurology, Johns Hopkins University School of Medicine, Baltimore, MD 21205, USA; mpolyde@jhmi.edu; 7Centre for Neuromuscular Diseases, Department of Neuromuscular Diseases, UCL Queen Square Institute of Neurology, London WC1N 3BG, UK; m.reilly@ucl.ac.uk; 8Department of Medicine (Neurology and Rheumatology), Shinshu University School of Medicine, Matsumoto 390-8621, Japan; sekijima@shinshu-u.ac.jp; 9Department of Rheumatology & Clinical Immunology, Amyloidosis Center of Expertise, University Medical Center Groningen, 9700 RB Groningen, The Netherlands

**Keywords:** systemic amyloidosis, hereditary transthyretin amyloid, immunoglobulin light chain amyloid, transthyretin gene-variant carrier, biomarker, neurofilament light chain, polyneuropathy, small fiber neuropathy, autonomic neuropathy

## Abstract

Peripheral and autonomic neuropathy are common disease manifestations in systemic amyloidosis. The neurofilament light chain (NfL), a neuron-specific biomarker, is released into the blood and cerebrospinal fluid after neuronal damage. There is a need for an early and sensitive blood biomarker for polyneuropathy, and this systematic review provides an overview on the value of NfL in the early detection of neuropathy, central nervous system involvement, the monitoring of neuropathy progression, and treatment effects in systemic amyloidosis. A literature search in PubMed, Embase, and Web of Science was performed on 14 February 2024 for studies investigating NfL levels in patients with systemic amyloidosis and transthyretin gene-variant (*TTR*v) carriers. Only studies containing original data were included. Included were thirteen full-text articles and five abstracts describing 1604 participants: 298 controls and 1306 *TTR*v carriers or patients with or without polyneuropathy. Patients with polyneuropathy demonstrated higher NfL levels compared to healthy controls and asymptomatic carriers. Disease onset was marked by rising NfL levels. Following the initiation of transthyretin gene-silencer treatment, NfL levels decreased and remained stable over an extended period. NfL is not an outcome biomarker, but an early and sensitive disease-process biomarker for neuropathy in systemic amyloidosis. Therefore, NfL has the potential to be used for the early detection of neuropathy, monitoring treatment effects, and monitoring disease progression in patients with systemic amyloidosis.

## 1. Introduction

Systemic amyloidoses are protein-misfolding diseases in which proteins adopt a misfolded state and aggregate into amyloid fibrils that subsequently deposit in the extracellular space of tissues and organs, leading to a progressive loss of function of the affected organs. The initially insidious course of these progressive and lethal diseases makes early detection difficult. Biomarkers for early detection are extremely useful because starting early with treatment hugely changes the dismal prospects of the patient [1,2].

Transthyretin (ATTR) amyloidosis and immunoglobulin light chain (AL) amyloidosis are the two main types of systemic amyloidosis that can affect the nervous system [1]. ATTR amyloidosis can be hereditary (ATTRv), the result of the deposition of variant transthyretin (TTRv), or acquired (ATTRwt), the result of the deposition of wild-type TTR [1]. The peripheral nervous system is frequently affected in ATTRv and AL amyloidosis, leading to polyneuropathy and autonomic neuropathy [2,3]. However, leptomeningeal involvement can occur in ATTRv amyloidosis [4], and peripheral polyneuropathy may also occur in ATTRwt amyloidosis [5].

Establishing the presence of polyneuropathy in patients with systemic amyloidosis is crucial for early diagnosis and the initiation of treatment. There are several treatment options for ATTR amyloidosis that either stabilize the TTR tetramer (TTR-stabilizers) to prevent its dissociation into amyloidogenic monomers [6,7] or ribonucleic acid interference treatments for controlling TTR-gene expression (*TTR*-gene silencers) [8,9]. AL amyloidosis is treated with chemo-immunotherapy to destroy the malignant plasma cells that produce the amyloidogenic free light chains [10]. Treatments in systemic amyloidosis are most beneficial when initiated in an early disease stage. The presence and severity of polyneuropathy have prognostic implications for survival and quality of life. In addition, as polyneuropathy will progress over time, monitoring its course is important for assessing disease progression and treatment effect.

Polyneuropathy is confirmed by nerve conduction studies (NCSs) showing axonal degeneration [11,12]. Although NCSs are the most objective measure for the evaluation of polyneuropathy, NCSs have limited sensitivity for axonal damage in early disease stages and are only able to measure large-fiber neuropathy [11,13]. Small-fiber neuropathy can be evaluated by quantitative sensory testing (QST). However, this non-invasive method is limited by its subjective nature [11,14]. The Sudoscan is another objective modality in the evaluation of small-fiber neuropathy but is limited to the sympathetic C nerve fibers of the autonomic nervous system [15]. Another option to evaluate nerve involvement in systemic amyloidosis is to perform a sural nerve biopsy. However, this procedure carries the risk of permanent cutaneous anesthesia in the biopsied nerve area [16]. Lastly, a punch skin biopsy can be performed to detect amyloid and small nerve fiber loss [17,18,19].

In addition, the neuropathy impairment score (NIS), neuropathy impairment score for the lower limbs (NIS-LL), neuropathy impairment score for the upper limbs (NIS-UL), modified neuropathy impairment score + 7 (mNIS + 7) (includes NCS, QST, and autonomic endpoints), familial amyloidotic polyneuropathy (FAP) stage, and polyneuropathy disability (PND) score were designated to assess polyneuropathy impairment in ATTRv amyloidosis [20]. As all these measures mentioned above assess damage to nerves and related nerve dysfunction, they all can be considered outcome markers. Outcome markers that are insensitive to tracking disease progression over shorter time intervals (i.e., within one year) and, because the best outcome will not be a clear improvement but merely stabilization, they do not quickly provide useful information about a favorable treatment effect. This is in sharp contrast to a disease-process biomarker that reflects the activity of the disease leading to the outcome. The earlier such a disease-process biomarker improves or even normalizes (i.e., within six months), the less damage will occur. Therefore, obtaining a disease-process biomarker would signify a major gain in the neuropathy toolbox of the clinician.

Biomarkers are available and very helpful as disease-process parameters for the detection and follow-up of cardiac disease (troponin T and N-terminal pro-brain-type natriuretic peptide (NT-proBNP)), liver disease (alkaline phosphatase, gamma-glutamyl transferase, and bilirubin), and kidney disease (urea, creatinine, proteinuria, and cystatin C) in amyloidosis [2,21]. Therefore, there is a clear need for an early and sensitive serum or plasma biomarker for polyneuropathy, for neuropathy progression and for assessing the effect of treatment on neuropathy in systemic amyloidosis. Recent research shows that the neurofilament light chain (NfL), a neuron specific cytoskeletal protein released into the blood and cerebrospinal fluid during axonal damage [22], correlates with polyneuropathy and disease severity in systemic amyloidosis [23,24,25]. What is more, NfL levels normalize after treating vasculitis as cause of polyneuropathy, traumatic brain injury, and stroke [26,27,28]. NfL thus has potential to behave as a disease-process marker not only for disease progression, but also signifying a favorable treatment effect on polyneuropathy. So far, measurements of NfL have been limited in amyloidosis to research settings, despite adequate studies exploring its potential value in systemic amyloidosis. NfL measurement is implemented in other diseases to monitor their courses, e.g., in Multiple Sclerosis [29]. This review aims to assess the current status of NfL for neuropathy in amyloidosis.

Various assays have been used to measure NfL levels. The initial assay developed for NfL measurement was an enzyme-linked immunosorbent assay (ELISA) using monoclonal antibodies directed against the conserved rod domain of NfL [30]. The ELISA is sufficient to measure NfL levels in cerebrospinal fluid but is not sensitive enough to quantify NfL levels in blood. However, more sensitive bioassays have been developed, such as electro chemiluminescent and enzymatic chemiluminescent assays. These assays make use of the same monoclonal antibodies but have a higher sensitivity than the standard ELISA assay. The fluorescence-based immunoassay, known as the single-molecule array (Simoa) assay, offers enhanced sensitivity, capable of detecting extremely low antigen concentrations in biofluids. The Simoa assay is currently the gold standard for measuring NfL in serum or plasma. Two newer assays with a lower limit of quantitation, low enough to be used for NfL measurement in serum or plasma, are the Siemens CLIA assay and the Simple Plex Ella assay [31]. 

Concerning AL amyloidosis, little is known about the response to treatment in AL amyloidosis patients with polyneuropathy. NfL may serve as a valuable biomarker for assessing the presence of polyneuropathy. Treatment decisions can be informed by NfL results, as certain treatment options may be more advisable to avoid in the context of AL amyloidosis [32].

We performed a systematic search of current studies on NfL in systemic amyloidosis to evaluate the value of NfL in the early detection of neuronal damage and monitoring polyneuropathy progression and the treatment effect. Currently, NfL measurements in serum or plasma are only performed in research settings of neuropathy in amyloidosis. Therefore, we evaluated both the evidence and practical challenges to implementing blood-NfL measurement into clinical practice to aid in diagnostic investigations and treatment decisions regarding neuropathy in amyloidosis in the near future.

## 2. Methods

### 2.1. Eligibility Criteria

Inclusion criteria were clinical studies from all publication years and any country of patients diagnosed with systemic amyloidosis as well as carriers of a variant in the *TTR*-gene in which NfL levels were measured. 

Exclusion criteria were (1) articles in languages other than English and (2) studies concerning patients with Alzheimer’s disease. Studies on Alzheimer’s disease were excluded because Alzheimer’s disease is regarded as an localized form of amyloidosis. This review is primarily focused on NfL in systemic amyloidosis. 

### 2.2. Information Sources

PubMed, Embase, and Web of Science were used.

### 2.3. Search Strategy

The following search query was used in PubMed, Embase, and Web of Science: ((((amyloidosis) OR (amyloid neuropathy)) NOT (Alzheimer)) AND (neurofilament) OR (NfL)). The references of the included articles were screened to find more articles.

### 2.4. Selection Process

A total number of 173 articles was found (PubMed: 58, Embase: 69, and Web of Science: 46) on the search date 14 February 2024. See Figure 1.

First, 62 duplicates were removed. Of the remaining 111 unique titles, 77 were excluded because the articles were written in languages other than English (*n* = 1), concerned animal studies (*n* = 10), or deemed irrelevant (*n* = 66), leaving 34 titles. After reading the abstracts, 5 additional publications were excluded because they were deemed irrelevant (*n* = 5), leaving 29 publications. Of these 29 publications, two were deemed irrelevant (*n* = 2), lacked original data (*n* = 7), or were written in a language other than English (*n* = 2), leaving 13 articles and 5 relevant congress abstracts.

No new articles could be added after checking the reference lists of the publications.

### 2.5. Data-Collection Process

Publications were categorized according to these topics in patients with amyloidosis: neurofilament light chain levels in relation to polyneuropathy and disease severity; neurofilament light chain levels in an asymptomatic disease stage; neurofilament light chain levels in relation to small-fiber and autonomic neuropathy; neurofilament light chain levels in relation to treatment; and confounders affecting neurofilament light chain levels.

### 2.6. Description of Cases

As not all cases will develop symptoms, cases can only in retrospect be described as presymptomatic after they have developed symptoms. Therefore, we chose to describe cases as asymptomatic instead of presymptomatic in this review, because asymptomatic describes the actual situation at the moment of evaluation.

In this review, we use the following terminology:Asymptomatic *TTR*v carriers: carriers of a pathogenic *TTR*-gene variant without symptoms or signs of polyneuropathy;Asymptomatic ATTRv amyloidosis patients: carriers of a pathogenic *TTR* variant without symptoms or signs of polyneuropathy, no signs of polyneuropathy on NCS, and no cardiomyopathy, but with amyloid detected in subcutaneous abdominal fat tissue or elsewhere;Symptomatic ATTRv amyloidosis patients: carriers of a pathogenic *TTR*-gene variant with symptoms or signs of polyneuropathy.

## 3. Results

### 3.1. Results of Individual Studies

Our systematic review comprised a total of 1604 participants, including 1286 ATTRv amyloidosis and 20 AL amyloidosis patients with and without neurological symptoms along with 298 healthy controls. Currently, there are no published studies on NfL in ATTRwt amyloidosis. Table 1 provides a summary of the 18 studies that were included. Figure 2 displays the NfL levels per study and per patient group.

**Table 1 ijms-25-03770-t001:** Study overview: neurofilament light chain levels and correlations with disease characteristics.

Study (Ref.)	Comparisons between Groups	Number of Subjects	Assay/Sample Type	Fold Increase in Median NfL	NfL and Correlation with Disease Characteristics	NfL and No Correlation with Disease Characteristics
**Full-text articles**						
Kapoor et al., 2019 [23]	Healthy controls vs. ATTRv no neuropathy	16 6	Simoa/plasma	0.2 (15.5 vs. 2.5) *	NIS scale, CMTES-R	
	Healthy controls vs. ATTRv-PNP	16 20		4.4 (15.5 vs. 68.4)		
	ATTRv no neuropathy vs. ATTRv-PNP	6 20		27.4 (2.5 vs. 68.4) *		
Loser et al., 2022 [24]	*TTR*v carriers vs. ATTRv-PNP	6 14	Simoa/serum	B: 3.6 (5.4 vs. 19.7) FU 1 year: 3.7 (7.5 vs. 28.0)	B and t1: PND score, FAP stage, R-ODS, SFN-SIQ, Norfolk-QOL-DN, NIS, NIS-UL, NIS-LL, ESC feet, ESC hands, NCS motor sum score, NCS sensory sum score.	CADT, handgrip right, handgrip left,
Louwsma et al., 2021 [25]	Healthy controls vs. *TTR*v carriers	15 15	Simoa/serum	0.8 (8.8 vs. 6.9)	PND score, sural nerve amplitude in ATTRv patients, troponin T in ATTRv patients with PNP	Sural nerve amplitude in *TTR*v carriers, digit 5 ulnar nerve amplitude, NT-proBNP, creatinine
	Healthy controls vs. ATTRv-PNP	15 15		7.5 (8.8 vs. 66.4)		
	*TTR*v carriers vs. ATTRv-PNP	15 15		9.6 (6.9 vs. 66.4)		
	ATTRv-PNP PND I vs. ATTRv-PNP PND ≥I	15 15		5.6 (21 vs. 116)		
	Healthy controls vs. AL no neuropathy	10 10		1.7 (13.6 vs. 22.7)	Troponin T in AL patients with and without PNP	NT-proBNP, creatinine
	Healthy controls vs. AL-PNP	10 10		11 (13.6 vs. 149)		
	AL no neuropathy vs. AL-PNP	10 10		6.6 (22.7 vs. 149)		
Maia et al., 2020 [33]	Healthy controls vs. *TTR*v carriers	16 16	Simoa/plasma	-	PND score	
	*TTR*v carriers vs. ATTRv-PNP	16 16		-		
	Healthy controls vs. ATTRv-PNP PND I	16 13		4.8		
	Healthy controls vs. ATTRv-PNP PND ≥ II	16 13		15.4		
Ticau et al., 2021 [34]	Healthy controls vs. ATTRv-PNP (all) baseline	57 189	Simoa/plasma	4.3 (16.3 vs. 69.4) *	Change in mNIS + 7 after 18 months of treatment with patisiran	mNIS + 7 at baseline and PND score at baseline
	ATTRv-PNP patisiran 18 months vs. ATTRv-PNP placebo 18 months	111 47		2.0 (48.8 vs. 99.5) *		
	Healthy controls vs. ATTRv-PNP patisiran 18 months	57 111		3.0 (16.3 vs. 48.8) *		
	Healthy controls vs. ATTRv-PNP placebo 18 months	57 47		6.1 (16.3 vs. 99.5) *		
Luigetti et al., 2022 [35]	Healthy controls vs. *TTR*v carriers and ATTRv-PNP	26 17	Ella/serum	4.5 (18 vs. 81.8) *	NIS scale, Sudoscan values from feet, interventricular septum thickness, Norfolk QOL-DN	FAP stage, PND score, CADT
Sato et al., 2023 [36]	ATTRv-PNP tafamidis vs. ATTR-PNP patisiran one year	11 11	Simoa/serum	0.7 (106.4 vs. 72.6) *		NIS score one and two years after treatment switch
	ATTRv-PNP tafamidis vs. ATTR-PNP patisiran two years	8 8		0.6 (92.8 vs. 55.9) *		
Lau et al., 2023 [37]	Healthy controls vs. ATTRv no neuropathy	25 7	Simoa/plasma	0.8 (14.5 vs. 11.9)	Creatinine	NIS-LL subscore, NT-proBNP, troponin I
	Healthy controls vs. ATTRv-PNP	25 11		2.5 (14.5 vs. 35.9)		
	ATTRv no neuropathy vs. ATTRv-PNP	7 11		3.0 (11.9 vs. 35.9)		
	ATTRv no neuropathy vs. ATTRv-PNP	7 6		FU 4 years: 1.5		
Ticau et al., 2023 [38]	ATTRv-PNP baseline vs. ATTRv-PNP patisiran 52 months	111 87	Simoa/plasma	0.6 (72.0 vs. 44.1) *	Change in mNIS + 7 and Norfolk QOL-DN sustained after 24 months additional patisiran treatment	
	ATTRv-PNP patisiran Global OLE baseline vs. ATTRv-PNP patisiran 24 months Global OLE	111 87		0.9 (48.8 vs. 44.1) *		
	ATTRv-PNP patisiran 30 months vs. ATTRv-PNP placebo 18 months → 12 months patisiran Global OLE	76 28		1.3 (50.1 vs. 64.0) *		
	ATTRv-PNP patisiran 42 months vs. ATTRv-PNP placebo 18 months → 24 months patisiran Global OLE	87 24		1.0 (44.1 vs. 42.8) *		
	ATTRv-PNP baseline vs. ATTRv-PNP patisiran 18 months Phase II OLE	26 25		0.8 (32.9 vs. 26.1) *		
	ATTRv-PNP baseline vs. ATTRv-PNP patisiran 48 months Global OLE	26 23		0.7 (32.9 vs. 23.0) *		
Romano et al., 2024 [39]	Healthy controls vs. *TTR*v carriers	5 50	Ella/serum	0.7 (17.7 vs. 13.1)	PND score, NIS score, FAP stage	
	Healthy controls vs. ATTRv-PNP	5 61		4.2 (17.7 vs. 74.0)		
	*TTR*v carriers vs. ATTRv-PNP	50 61		5.6 (13.1 vs. 74.0)		
González-Moreno et al., 2024 [40]	Healthy controls vs. *TTR*v V30M carriers	30 31	ELISA/serum	Incalculable (<33 vs. <33)	NIS score	FAP stage
	Healthy controls vs. symptomatic ATTRv V30M	30 29		Incalculable (<33 vs. 116)		
	*TTR*v V30M carriers vs. symptomatic ATTRv V30M	31 29		Incalculable (<33 vs. 116)		
Carroll et al., 2024 [41]	Asymptomatic (PND 0) vs. symptomatic (PND ≥ I)	11 16	Simoa/serum	9.4 (14.3 vs. 134)	Baseline: PND score, FAP stage, NIS, NIS-LL, CMTSS, CMTES, CMTNS, MRC scores	eGFR, creatinine, Baseline: Norfolk-QOL-DN
Berends et al., 2024 [42]	*TTR*v carriers vs. ATTRv no neuropathy	12 8	Simoa/serum	0.9 (8.2 vs. 7.1)	PND score	
	*TTR*v carriers vs. ATTRv-PNP TTR-stabilizer	12 20		5.3 (8.2 vs. 43.2)		
	*TTR*v carriers vs. ATTRv-PNP patisiran	12 18		7.5 (8.2 vs. 61.2)		
	*TTR*v carriers and ATTRv no neuropathy vs. *TTR*v carrier who developed PNP baseline	20 7		1.1 (7.6 vs. 8.40)		
	ATTRv-PNP TTR-stabilizer Vs ATTRv-PNP patisiran	20 18		1.4 (43.2 vs. 61.2)		
	ATTRv-PNP TTR-stabilizer vs. *TTR*v carrier who developed PNP PND ≥ I	20 7		1.2 (43.2 vs. 49.8)		
	ATTRv-PNP patisiran vs. *TTR*v carrier who developed PNP PND ≥ I	18 7		0.8 (61.2 vs. 49.8)		
**Abstracts**						
Ticau et al., 2020 [43]	Healthy controls vs. ATTRv-CM no neuropathy	53 93	Not specified/plasma	3.3 (16.3 vs. 54.1) *	PND score	Cardiomyopathy
	Healthy controls vs. ATTRv-CM PND >0	53 101		3.8 (16.3 vs. 61.4) *		
	Healthy controls vs. ATTRv-PNP APOLLO	53 193		4.3 (16.3 vs. 69.4) *		
	ATTRv-CM no neuropathy vs. ATTRv-CM PND >0	93 101		1.3 (46.2 vs. 61.4) *		
	ATTRv-CM no neuropathy vs. ATTRv-PNP APOLLO	93 193		1.5 (46.2 vs. 69.4) *		
	ATTRv-CM PND >0 vs. ATTRv-PNP APOLLO	101 193		1.1 (61.4 vs. 69.4) *		
Berends et al., 2022 [44]	[^123^I]*m*IBG-scintigraphy negative *TTR*v carriers and ATTRv patients vs. [^123^I]*m*IBG-scintigraphy positive *TTR*v carriers and ATTRv patients	22 16	Simoa/serum	4.8 (9.2 vs. 44.0)	NCS, PND score, NT-proBNP, troponin T, late heart-to-mediastinum ratio, wash-out rate, Ewing battery tests, [^123^I]*m*IBG-scintigraphy	
Conçeicao et al., 2023 [45]	ATTRv-PNP eplontersen vs. ATTRv-PNP inotersen until week 35 followed by eplontersen	144 24	Ella/serum			
Luigetti et al., 2023 [46]	ATTRv-PNP patisiran baseline vs. ATTRv-PNP patisiran 4 months	36 36	Simoa/plasma	0.8 (55.7 vs. 46.0) *		
	ATTRv-PNP patisiran baseline vs. ATTRv-PNP patisiran 18 months	36 36		0.7 (55.7 vs. 39.3) *		
	ATTRv-PNP vutrisiran baseline vs. ATTRv-PNP vutrisiran 4 months	111 111		0.8 (59.1 vs. 48.1) *		
	ATTRv-PNP vutrisiran baseline vs. ATTRv-PNP vutrisiran 18 months	111 111		0.7 (59.1 vs. 39.2) *		
	ATTRv-PNP patisiran baseline vs. ATTRv-PNP vutrisiran baseline	36 111		1.1 (55.7 vs. 59.1) *		
	ATTRv-PNP patisiran 18 months vs. ATTRv-PNP vutrisiran 18 months	36 111		1.0 (39.3 vs. 39.2) *		
Gilling et al., 2023 [47]	ATTRv-PNP placebo baseline vs. ATTRv-PNP placebo → patisiran 36 months	47 15	Simoa/plasma	(63.2 vs. 40.0) *		
	ATTRv-PNP patisiran baseline vs. ATTRv-PNP patisiran 18 months + Global OLE patisiran 36 months	111 72		(72.0 vs. 44.8) *		
	ATTRv-PNP Phase II OLE patisiran 24 months + Global OLE 36 months	19		26.1 *		

AL: immunoglobulin light chain amyloid; ATTRv: hereditary transthyretin amyloid; CADT: compound autonomic dysfunction test; CM: cardiomyopathy; CMTES: Charcot–Marie–Tooth symptom and examination subscore; CMTNS: Charcot–Marie–Tooth Neuropathy Score version 2; CMTSS: Charcot–Marie–Tooth symptom subscore; EMG: electromyography; ESC: electrochemical skin conductance; FAP: Familial amyloid polyneuropathy; mNIS + 7: modified Neuropathy Impairment Score + 7; MRC: Medical Research Council power score; NCS: nerve-conduction studies; NIS: Neuropathy Impairment Score; NIS-LL: Neuropathy Impairment score–lower limbs; NIS-UL: Neuropathy Impairment score–upper limbs; Norfolk QOL-DN: Norfolk quality of life diabetic neuropathy; NT-proBNP: N-terminal pro-brain-type natriuretic peptide; [^123^I]*m*IBG-scintigraphy: iodine-123 labelled meta-iodobenzylguanidine scintigraphy; OLE: open-label extension; PND: Polyneuropathy Disability; PNP: polyneuropathy; R-ODS: Rasch-built Overall Disability Score; SFN-SIQ: Small fiber Neuropathy–Symptom Inventory Questionnaire; t1: first timepoint of follow-up; *TTR*v: transthyretin gene variant; and V30M: TTRVal30Met p.(Val50Met). * Data are expressed as mean values.

**Figure 2 ijms-25-03770-f002:**
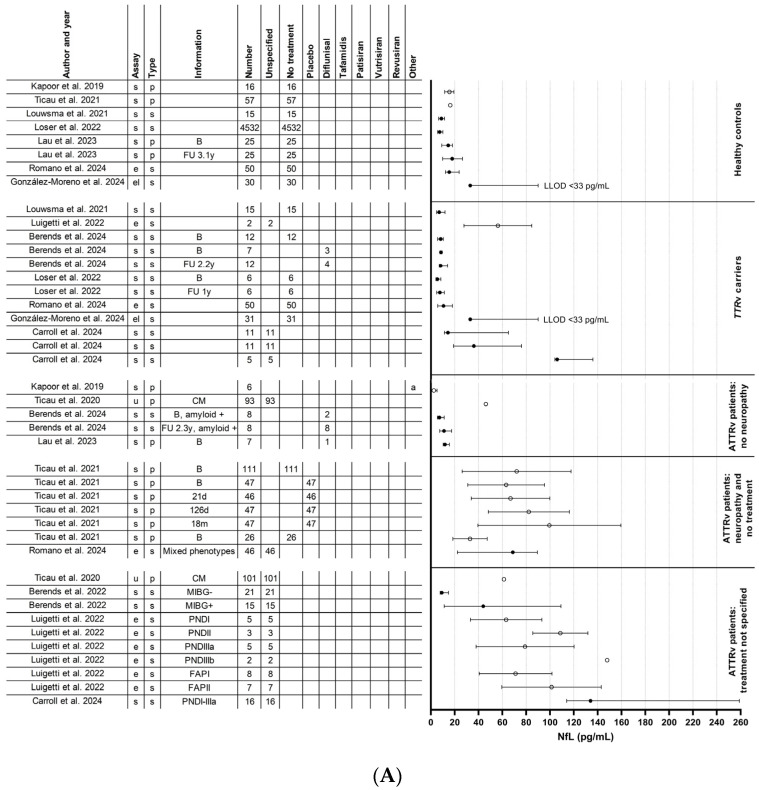
Neurofilament light chain levels in all studies. (**A**) Neurofilament light chain levels in healthy controls, *TTR*v carriers, ATTRv patients without neuropathy, and ATTRv patients with neuropathy with or without treatment. (**B**) Neurofilament light chain levels in ATTRv patients with neuropathy and with treatment. Solid circles represent studies with median and interquartile ranges, open circles represent studies with means and standard deviations. Berends et al., 2022 [44], Berends et al., 2024 [42], Carroll et al., 2024 [41], Gilling et al., 2023 [47], González-Moreno et al., 2024 [40], Kapoor et al., 2019 [23], Lau et al., 2023 [37], Loser et al., 2022 [24], Louwsma et al., 2021 [25], Luigetti et al., 2022 [35], Luigetti et al., 2023 [46], Romano et al., 2024 [39], Sato et al., 2023 [36], Ticau et al., 2020 [43], Ticau et al., 2021 [34], Ticau et al., 2023 [38]. (a) Five patients used diflunisal and one patient used tafamidis, but it was not specified to which study group these patients belonged; (b) one patient used diflunisal but switched to tafamidis during the follow-up period.; (c) two patients used inotersen; (d) patients used a TTR-stabilizer or a TTR-gene silencer, but it was unspecified which one; (e) six of the eight untreated patients had received a liver transplantation in the past and two of the four patients treated with patisiran had received a liver transplantation in the past; (f) all (six) untreated patients had received a liver transplantation in the past and two of the six patients treated with patisiran had received a liver transplantation in the past; (g) one patient used diflunisal and switched to tafamidis during the follow-up period and one patient used diflunisal and switched to eplontersen during the follow-up period. ATTRv: hereditary transthyretin amyloid; B: baseline; CM: cardiomyopathy; e: Ella; el: ELISA; FAP: familial amyloid polyneuropathy; FU: follow-up; m: month; LLOD: lower limit of detection; MIBG: meta-iodobenzylguanidine scintigraphy; NfL: neurofilament light chain; p; plasma; PND: polyneuropathy disability; PNP: polyneuropathy; s: serum; s: single-molecule array (Simoa); TTRv: transthyretin gene variant; and y: year.

### 3.2. Neurofilament Light Chain in Relation to Polyneuropathy and Disease Severity

Nine studies compared serum or plasma NfL levels in ATTRv amyloidosis patients with polyneuropathy to concentrations in neurologically asymptomatic *TTR*v carriers, asymptomatic ATTRv patients, or healthy controls [23,24,25,33,34,35,39,40,41]. One of these studies also compared serum NfL levels in AL amyloidosis patients with polyneuropathy to levels in AL amyloidosis without polyneuropathy or healthy controls [25]. All these studies found increased levels in patients with polyneuropathy compared to asymptomatic *TTR*v carriers, neurologically asymptomatic ATTRv patients or healthy controls. Details of these studies are shown in Table 1 and Figure 2.

Six studies found a correlation between NfL levels and disease severity measured by the (modified + 7) Neuropathy Impairment Score (NIS) [23,24,35,39,40,41], six studies between NfL levels and disease severity measured by the polyneuropathy disability (PND) score [24,25,33,39,41,42], and three studies between NfL and the FAP stage [24,39,41]. However, one of the largest studies found no correlation between mNIS + 7 score or PND score and plasma NfL levels [34]. Other studies also found no correlations between PND score or FAP stage and serum NfL [35,40] or NIS-LL scores and plasma NfL [37]. One study found that changes in plasma NfL correlated with mNIS + 7 during treatment with patisiran [34], and another study showed that a sustained improvement in mNIS + 7 score goes in parallel with maintained reduced plasma NfL levels after treatment with patisiran [38]. In contrast, Sato et al. observed a significant decrease in serum NfL levels at one and two years after the switch to patisiran, whereas NIS scores did not change [36].

### 3.3. Neurofilament Light Chain in an Asymptomatic Disease Stage

Six studies [24,33,34,39,40,41] conducted receiver operating characteristics (ROC) analysis to assess the ability of NfL to distinguish between an asymptomatic stage and a symptomatic neuropathy stage. Details of these studies are shown in Table 1 and Table 2 and Figure 2.

First, Ticau et al. [34] discriminated healthy controls and ATTRv amyloidosis patients with polyneuropathy based on a plasma NfL cutoff level of 37 pg/mL measured with the Simoa assay with a sensitivity of 84.9% and a specificity of 96.4%. Second, Romano et al. [39] established a serum NfL cutoff, which was almost the same as from Ticau et al., of 37.10 pg/mL for the transition from asymptomatic to symptomatic, with a sensitivity of 81.4% and a specificity of 100% with the Ella assay. However, they used different analytical methods and sample types (serum or plasma). Third, Loser et al. [24] concluded that serum NfL levels above 11.7 pg/mL measured with a Simoa assay at both baseline and after 1-year follow-up could discriminate symptomatic from asymptomatic patients with 85.7% sensitivity and 100% specificity. No significant difference was found between healthy controls and asymptomatic ATTRv amyloidosis patients. Fourth, Maia et al. [33] studied plasma NfL levels in two independent cohorts with a Simoa assay. A NfL cutoff of 10.6 pg/mL discriminated between asymptomatic (PND 0) and early-stage patients (PND I) and between asymptomatic (PND 0) and symptomatic patients (PND ≥ I) with a sensitivity of 92.3% and 96.2%, respectively, and a specificity of 93.8% in both. In addition, comparing PND I with PND ≥ II in cohort #1 resulted in an optimal cutoff of 66.9 pg/mL with a sensitivity of 61.5% and a specificity of 92.3%, whereas the optimal cutoff was set on 75.7 pg/mL in cohort #2 with a sensitivity of 84.6% and a specificity of 80%. Fifth, González-Moreno et al. [40] established that a cutoff value of 93.6 pg/mL discriminates ATTRv amyloidosis patients from asymptomatic *TTR*v carriers with a sensitivity of 79% and a specificity of 87%. The high cutoff could be related to their use of a first-generation ELISA to measure serum NfL levels. The NfL levels they found are higher compared to levels reported in studies using the Ella or Simoa assay. Sixth, Carroll et al. [41] found that baseline NfL levels greater than 64.5 pg/mL discriminated between a combined group of symptomatic patients and individuals who were at baseline asymptomatic but developed sensorimotor neuropathy (sensorimotor converters), and asymptomatic individuals with a sensitivity of 91.9% and a specificity of 88.5%. Asymptomatic individuals could only be discriminated from a combined group of sensory and sensorimotor converters or symptomatic patients by NfL levels above 88.9 pg/mL with a sensitivity 62.9% and a specificity of 96.2%. However, an increase of 17% in NfL levels over 6 months could discriminate asymptomatic from sensory or sensorimotor converters with a sensitivity of 88.9% and a specificity 80.0%. 

Lau et al. [37] followed six initially neurologically asymptomatic patients who developed polyneuropathy. During the study period, plasma NfL levels increased 1.5-fold from baseline over a follow-up period of 4.0 years [3.6–6.9]. They did not establish cutoff values for discriminating between no polyneuropathy and polyneuropathy. Their cohort, with only V122I-genopositive patients, likely had mild or subtle polyneuropathy, making a cutoff value hard to find due to small differences between asymptomatic and symptomatic patients in their study. 

In addition, Berends et al. [42] studied serum NfL levels longitudinally in twelve asymptomatic *TTR*v carriers and eight asymptomatic ATTRv patients. Serum NfL increased over two years in asymptomatic ATTRv amyloidosis patients, but did not change in the asymptomatic *TTR*v carriers. Levels of serum NfL were also studied longitudinally in a group of seven *TTR*v carriers who progressed during a median follow-up of ten years from asymptomatic *TTR*v carriers to symptomatic ATTRv amyloidosis patients with NCS-confirmed polyneuropathy. In this group of *TTR*v carriers who developed polyneuropathy, the median baseline serum NfL level of 8.4 pg/mL rose to a median of 49.8 pg/mL upon the onset of initial symptoms (PND I), and the serum NfL level had risen even further at the time polyneuropathy could be established by nerve-conduction studies. Levels of serum NfL were already above the 95th reference percentile 5.5 years (range 3.0–7.6 years) before the onset of symptoms (PND I).

### 3.4. Neurofilament Light Chain in Relation to Small-Fiber and Autonomic Neuropathy

The axonal length-dependent polyneuropathy in amyloidosis is typically preceded by a small-fiber neuropathy in the lower extremities [48,49]. Detecting small-fiber neuropathy is thus desirable for early disease detection. The Sudoscan serves as an objective test for evaluating small-fiber neuropathy limited to the sympathetic C nerve fibers of the autonomic nervous system [15]. In a study by Luigetti et al. [35], a significant association was demonstrated between serum NfL values and Sudoscan values obtained from the feet. However, the participants in this study also exhibited (high-grade) polyneuropathy, which confounded this correlation. Therefore, at present, it remains unknown whether NfL detects small-fiber neuropathy in the absence of polyneuropathy. Details of this study are shown in Table 1 and Figure 2.

Autonomic dysfunction is also a common and early manifestation in ATTRv amyloidosis. Data on serum NfL levels in relation to cardiac autonomic neuropathy based on iodine-123-labeled meta-iodobenzylguanidine ([^123^I]*m*IBG) scintigraphy and other measures of autonomic neuropathy (Ewing Battery) have been presented [44]. In multivariate regression analysis, polyneuropathy was the only independent predictor of serum NfL levels, in contrast to (cardiac) autonomic neuropathy. Details of this study are shown in Table 1 and Figure 2.

### 3.5. Neurofilament Light Chain in Relation to Amyloid-Specific Treatment

Ten studies [24,34,36,37,38,41,42,45,46,47] also analyzed the effect of treatment with a TTR-stabilizer or *TTR*-gene silencer which was given to retard or halt the disease progression of the amyloidosis. Details of these studies are shown in Table 1 and Table 3 and Figure 2.

Ticau et al. [34] analyzed plasma NfL levels in a subset of ATTRv amyloidosis patients who participated in the phase 3, placebo-controlled study of patisiran (APOLLO-A). Patients treated with patisiran showed a significant decrease in plasma NfL levels at 9 months compared to baseline, which was maintained at 18 months. In contrast, patients in the placebo group showed an increase in plasma NfL levels at 9 months compared to baseline, and this increase was maintained at 18 months. At the 18-month mark, patients receiving patisiran exhibited plasma NfL levels that were twice as low as those without treatment (placebo group). Additionally, after 18 months of patisiran treatment, mNIS + 7 scores improved, and this improvement correlated with a decrease in plasma NfL levels. However, no correlation was observed between plasma NfL levels and mNIS + 7 or PND score at baseline.

The patisiran Global open-label extension (OLE) study [38] revealed that the reduced levels of plasma NfL, along with improvements in clinical efficacy assessments, persisted for an additional period of 24 months. Unpublished data by Gilling [47] again showed the maintained reduction in plasma NfL levels at the 36-month mark of the open-label extension study. Patients who received a placebo for 18 months in the APOLLO-A study [34] were switched to patisiran in the Global OLE study [38]. After 12 and 24 months of treatment with patisiran, plasma NfL levels decreased compared to the APOLLO-A baseline and even significantly decreased compared to the Global OLE baseline. After 24 months, these patients reached plasma NfL levels that were comparable to plasma NfL levels in the APOLLO-A patisiran group at 24 months in the Global OLE study. Data showed that this reduction persisted after 36 months (unpublished) [47].

Loser et al. [24] studied serum NfL levels in a cohort of patients who were untreated but had previously received a liver transplant or were under treatment with tafamidis or patisiran. Serum NfL levels tended to increase during one-year follow-up in untreated symptomatic patients, all of whom had received a liver transplant in the past. Serum NfL levels in patients on treatment with either patisiran (*n* = 4) or tafamidis (*n* = 2) also showed a tendency to increase during one year of follow-up. In patients initiated on patisiran (*n* = 2) during follow-up, serum NfL levels showed a tendency to decrease. All the patients in this study did not worsen neurologically during the follow-up period despite increasing serum NfL levels in some patients.

Sato et al. [36] longitudinally analyzed changes in serum NfL levels of patients who switched from tafamidis to patisiran. They observed a significant reduction in serum NfL levels one year after switching to patisiran, which was maintained over two years. These findings are in line with the findings of Loser et al. and Ticau et al. [24,34,38]. Interestingly, there was no significant change in NIS scores during the same time period.

Lau et al. [37] observed plasma NfL levels in patients who had polyneuropathy at baseline or developed it during follow-up. At the end of the follow-up period, some of the patients with polyneuropathy were treated with diflunisal, tafamidis, eplontersen, patisiran, or revusiran. No sustained plasma NfL level changes were observed with treatment initiation or regimen changes, and NIS scores did not correlate meaningfully with plasma NfL fluctuations.

Carroll et al. [41] longitudinally evaluated the effect of *TTR*-gene-silencer treatment on NfL levels in thirteen ATTRv amyloidosis patients. Levels of NfL decreased during treatment and the change in NfL levels positively correlated with the change in transthyretin levels over the same time interval.

Data on plasma NfL levels after 4 and 18 months after treatment initiation with patisiran or vutrisiran have been presented. Plasma NfL levels decreased at 4 months relative to baseline, and this decrease was sustained at 18 months for both treatment regimens [46].

Data on serum NfL levels in patients with ATTRv amyloidosis and Coutinho Stage 1-2 polyneuropathy that were treated with eplontersen have also been presented. Patients receiving eplontersen throughout week 85 showed a trend of decreasing serum NfL levels [45].

Except for the study conducted by Loser [24], the studies mentioned above show that NfL levels decrease, but do not normalize, after initiation of treatment with a *TTR*-gene silencer. A decrease in NfL levels within four months after treatment initiation with the *TTR*-gene silencer patisiran is maintained for at least 36 months. Two studies included some patients in whom serum or plasma NfL levels were studied after the initiation of treatment with a TTR-stabilizer [37,42], but there are insufficient data to draw conclusions about the effect of the initiation of TTR-stabilizers on serum or plasma NfL levels. In patients already treated with a TTR-stabilizer before baseline, serum NfL levels remained stable after two years of follow-up [42]. Due to the limited number of patients undergoing TTR-stabilizer treatment in the other studies, these studies are not suitable for drawing conclusions on this matter.

### 3.6. Neurofilament Light Chain and Cerebral Manifestations in Hereditary ATTR Amyloidosis

NfL is a biomarker of axonal damage of both the central and peripheral nervous system. Increased blood levels of NfL have been reported in almost all neurodegenerative disorders, among which are sporadic (amyloid-beta) cerebral amyloid angiopathy and Alzheimer’s disease [22,50]. Cerebral involvement in ATTRv amyloidosis shows many neuropathological and imaging similarities with sporadic cerebral amyloid angiopathy [51,52]. It is likely that (subclinical) cerebral involvement in ATTRv amyloidosis causes increased blood levels of NfL. That cerebral involvement in ATTRv amyloidosis can potentially lead to increased NfL levels has been mentioned in several studies [33,38,39], but no studies have actually investigated this. 

### 3.7. Confounders Affecting Neurofilament Light Chain Levels

Several influencing factors should be considered for the accurate interpretation of NfL levels in patients with ATTRv amyloidosis.

First, NfL is not specific to ATTRv amyloidosis-related polyneuropathy. Any cause of neuronal damage, whether cerebral or peripheral, may result in elevated levels of NfL [22]. Second, NfL levels increase with age [53]. NfL levels are expected to increase by 2.1% per year [54]. In individuals aged 60 years and older, there is an increase in the variability of NfL levels, possibly associated with subclinical comorbid pathology [54]. Eight studies included in this review took into account the effect of aging on NfL levels [23,24,25,33,37,39,40,41]. However, in all these studies, the NfL increase due to polyneuropathy outweighed the increase associated with aging. Third, both serum creatinine and hemoglobin A1c (HbA1c) exhibit strong correlations with NfL levels, even after adjusting for age. Kidney function plays a crucial role in NfL clearance, and patients with elevated HbA1c levels may experience microvascular disease complications leading to NfL release [54]. Fourth, body mass index (BMI) can affect NfL levels. Individuals with a higher BMI have a larger volume of distribution leading to lower absolute NfL levels [55]. A study involving 1706 individuals without neurological disease, which assessed the predictive capacity of 52 demographic, lifestyle, comorbidity, anthropometric, or laboratory characteristics in explaining variability in serum NfL levels, did not identify additional independent predictors [54].

## 4. Discussion

The primary objective of this systematic review was to ascertain the value of NfL in the early detection of neuropathy and the monitoring of neuropathy progression and treatment effect in systemic amyloidosis. In addition, this review aimed to assess the feasibility of implementing NfL in clinical practice in the near future. There is substantial evidence for the use of NfL as marker of polyneuropathy and neuropathy severity in ATTRv amyloidosis. There is also substantial evidence supporting the use of NfL in monitoring disease progression and the treatment effect of *TTR*-gene silencers. Some evidence supports the use of NfL in detecting neuropathy in a presymptomatic stage. However, in this context it is important to take into account that some evidence suggests that NfL is not suitable to detect small-fiber neuropathy [56] and autonomic neuropathy [44]. Only one study evaluated NfL in AL amyloidosis [29], and no studies have been published on ATTRwt amyloidosis. 

All available studies (Table 1) consistently show that NfL levels are increased in patients with ATTRv amyloidosis and polyneuropathy. The median levels are 4.3 to 15.4 times higher in patients with polyneuropathy compared to healthy controls, depending on the disease severity. The levels of NfL in patients with ATTRv amyloidosis with polyneuropathy are even higher than those observed for other peripheral nerve disorders, like chronic inflammatory demyelinating neuropathy [57] and Charcot–Marie–Tooth disease [58]. As ATTRv amyloidosis is a relatively rapidly progressive disease and, without treatment, fatal 7–12 years after the first disease manifestation [59], it can be hypothesized that this rate of progression contributes to higher NfL levels [23] even when the NIS score is lower than in, e.g., Charcot–Marie–Tooth disease. 

Based on the combined NfL data from the current studies, we could construct a hypothetical course of NfL levels over time from asymptomatic *TTR*v carriers who progress to asymptomatic ATTRv amyloidosis patients without neurological symptoms to symptomatic ATTRv amyloidosis patients with polyneuropathy and who subsequently receive treatment (Figure 3). Initially, the course of NfL levels in asymptomatic *TTR*v carriers resembles that of a healthy person [23,24,25,33,37,39,42]. However, when the amyloid is deposited, transforming the asymptomatic *TTR*v carrier into an asymptomatic ATTRv amyloidosis patient, NfL levels start to rise more than can be expected by aging alone [41,42]. In the subsequent period, the first clinical manifestations emerge, and polyneuropathy can be confirmed with NCS, and levels of NfL continue to rise [23,24,25,33,34,37,39,41,42]. During treatment with a TTR-stabilizer, NfL levels either remain stable [42] or may increase in individual patients [36], whereas levels of NfL decrease after the initiation of a *TTR*-gene silencer, and this decrease is sustained with extended treatment, up to 36 months, but levels do not normalize [34,36,38,41,45,46,47]. 

The lack of normalization in NfL, in contrast to what is observed in patients after the treatment of vasculitis as a cause of polyneuropathy, stroke, and traumatic brain injury [26,27,28], may have several explanations. First, despite the halt in disease progression, the axons already affected may gradually degenerate through a dying-back mechanism [60]. This smoldering axonal damage may cause an ongoing leakage of NfL from the neurons. Second, the existing amyloid deposits between axons may remain to be toxic to the neurons, subsequently leading to the continuous release of NfL [61]. Third, despite a significant reduction in TTR levels due to *TTR*-gene-silencer treatment, a residual quantity of TTR persists in the bloodstream which still could deposit on pre-existing amyloid deposits and consequently cause continuous, subtle nerve damage [62]. 

Apparently, conflicting results concerning a relationship between NfL levels and measures of disease severity have been reported. Several studies found correlations between NfL levels and disease severity measured by the different NIS scores (NIS, NIS-LL, mNIS + 7) [23,24,35,39,40,41], PND score and/or FAP stage [24,25,33,39,41,42], while one of the largest studies did not find a correlation between individual NfL levels and the mNIS + 7 score [34]. Sato et al. found that levels of NfL significantly decreased one and two years after the initiation of patisiran, with no change in NIS values [36]. In contrast, Ticau et al. reported a significant correlation between the decrease in NfL levels and an improvement in mNIS + 7 score after 18 months of treatment with patisiran [34]. The most likely explanation for these apparently contradictory results is that NfL reflects the active process of neuronal damage at a specific point in time, whereas PND and NIS scores reflect the overall burden of neurological impairment. PND and NIS scores can be considered outcome markers, whereas NfL is a disease-process biomarker. Consequently, it makes sense that these markers do not always correlate with each other.

Three different analytical technologies and either serum or plasma samples were used for measuring NfL in the studies included in this review. Three studies used the Ella assay [35,39,45], fourteen studies used the Simoa assay [23,24,25,33,34,35,36,37,38,42,43,44,46,47], and one study used a first-generation ELISA [40]. Both the Ella and Simoa assays make use of ultrasensitive immunoassay technology, and there is a good correlation between the outcomes of both assays. The Ella and Simoa assays use the same anti-NfL antibodies, but NfL levels are 17–24% higher when measured with the Ella assay compared to levels measured with the Simoa assay [31,63,64]. This absolute difference in concentration could be attributed to the use of different standards: Ella utilizes naturally derived bovine NfL, whereas Simoa utilizes recombinant human NfL [64]. The first-generation ELISA used in one of the studies included in this review was reported to have a lower limit of detection of 33 pg/mL [40]. Many of the asymptomatic *TTR*v carriers included in this study had NfL levels below the lower limit of detection. Therefore, this ELISA lacks the sensitivity needed to detect early neuronal damage in asymptomatic *TTR*v carriers that transition to symptomatic patients. Eight studies included in this review used plasma [23,33,34,37,38,43,46,47] and ten studies used serum [24,25,35,36,39,40,41,42,44,45]. There are proportional and systematic differences between serum and plasma NfL measurements. Plasma NfL levels are approximately 10% lower than serum NfL levels [65,66]; however, results can be used interchangeably if standardized values are used [67]. The pre-analytical stability of NfL is good: concentrations of NfL in serum or plasma remain stable at room temperature when the processing of samples is delayed up to 7 days [66], and concentrations of NfL remain stable in serum and plasma samples stored at −80 °C for up to 20 and 16 years, respectively [68].

### 4.1. Clinical Implications

Current evidence supports the implementation of NfL as early and sensitive serum or plasma biomarkers for polyneuropathy, for neuropathy progression, and for assessing the effect of treatment on neuropathy in ATTRv amyloidosis. 

NfL has added value compared to polyneuropathy impairment measures (e.g., FAP stage, PND, and NIS scores) and NCS, which can be considered outcome measures, as NfL is a biomarker for the neuropathic disease process. In addition, NfL measurement is not invasive, is easy to perform, requires only little time from the patient and healthcare provider, and is reproducible and objective. The established cutoffs to discriminate asymptomatic *TTR*v carriers from ATTRv amyloidosis patients with polyneuropathy vary considerably and depend on the sample type (serum versus plasma) and the assay that was used (Table 2). Some cutoffs have limited sensitivity and therefore cannot be used to rule out the presence of polyneuropathy; other cutoffs lack specificity (Table 2). Center-specific cutoff values may be useful but have to be established for each particular center. 

The best approach for incorporating NfL measurements into clinical practice seems to compare the measured NfL level with age-dependent reference values from one of two large online databases comprising individuals without a neurological disorder [53,69]. If the value exceeds the 95th percentile of normal for age, additional neurological examination and/or vigilance for the onset of polyneuropathy is recommended [42]. Another useful approach could be to look at changes in NfL levels over time (e.g., per three to six months) instead of absolute values at one moment. Carroll et al. [41] showed that a relative increase in NfL over time could discriminate asymptomatic *TTR*v carriers from carriers that developed sensory or sensorimotor neuropathy with good sensitivity. This approach provided better discrimination than assessing a single NfL value, in particular for the detection of neuronal damage in an early disease stage. This is also supported by the longitudinal data of Berends et al. [42]. It is relevant to detect neuronal damage, even in a presymptomatic stage, because treatment with diflunisal can then be considered [6].

NfL is sensitive to tracking disease progression and treatment effect over short time intervals, thus providing added value compared to disease-outcome measures in monitoring both disease progression and treatment effect. Increasing NfL levels indicate disease progression, while decreasing levels after the initiation of treatment indicate a beneficial treatment effect. NfL measurements will likely serve as useful adjunct measurements in future clinical trials.

When implementing NfL in daily practice, several factors need to be considered. First, NfL levels should be measured using a reliable and sensitive immunoassay, e.g., the Ella or Simoa assay, both of which unfortunately are not widely available. However, also routine laboratory technologies, such as Lumipulse [70], allow for the straightforward, reliable and sensitive longitudinal quantification of serum and plasma NfL. Confounders such as the presence of other neurological diseases, renal insufficiency, diabetes mellitus with microvascular complications, and aging should be taken into account when interpreting NfL levels. NfL is an early and sensitive marker for polyneuropathy, but there is some evidence that it does not detect small-fiber neuropathy and autonomic neuropathy. Therefore, NfL cannot be used as an absolute marker of neuropathy onset in ATTRv amyloidosis. 

### 4.2. Considerations for Future Research

NfL is a reliable and objective measure to detect neuronal damage in a presymptomatic stage in ATTRv amyloidosis [42]. A longitudinal investigation of a larger number of *TTR*v carriers with a variety of genotypes is warranted to specify the dynamics of NfL in *TTR*v carriers that progress to symptomatic ATTRv amyloidosis patients over time. There is a lack of studies investigating the use of NfL as a marker for small-fiber neuropathy, autonomic neuropathy, and central nervous system involvement in ATTRv amyloidosis. Studying NfL in relation to intra-epidermal nerve-fiber density (IENFD) would be of particular interest, as IENFD has shown to be a sensitive marker for the early detection of ATTRv amyloidosis [19]. NfL bears potential as a marker to detect polyneuropathy in patients with ATTRwt amyloidosis and ATTRv amyloidosis patients with apparently only cardiomyopathy [43]. However, studies in these patients are currently lacking. Another research gap that merits investigation pertains to the role of NfL in AL amyloidosis. Despite the higher occurrence of this form of systemic amyloidosis compared to ATTRv amyloidosis, only one study has examined NfL in AL amyloidosis.

### 4.3. Limitations

The major limitation of this systemic review on NfL in systemic amyloidosis is the lack of a meta-analysis. This is attributed to heterogeneity among the studies resulting from variations in NfL measurements (sample types and analytical methods), outcome variables, the composition of the study groups, and the composition of the control groups. Furthermore, the included studies did not all specify exactly how systemic amyloidosis was confirmed. Nevertheless, the results are very consistent across the various studies, allowing for clear conclusions to be drawn.

## 5. Conclusions

NfL is not an outcome biomarker, but an early and sensitive disease-process biomarker for neuropathy, particularly large-fiber neuropathy, in systemic amyloidosis. Therefore, NfL has potential to be used for the early detection of peripheral neuropathy and for monitoring treatment effects and disease progression in patients with systemic amyloidosis.

## Figures and Tables

**Figure 1 ijms-25-03770-f001:**
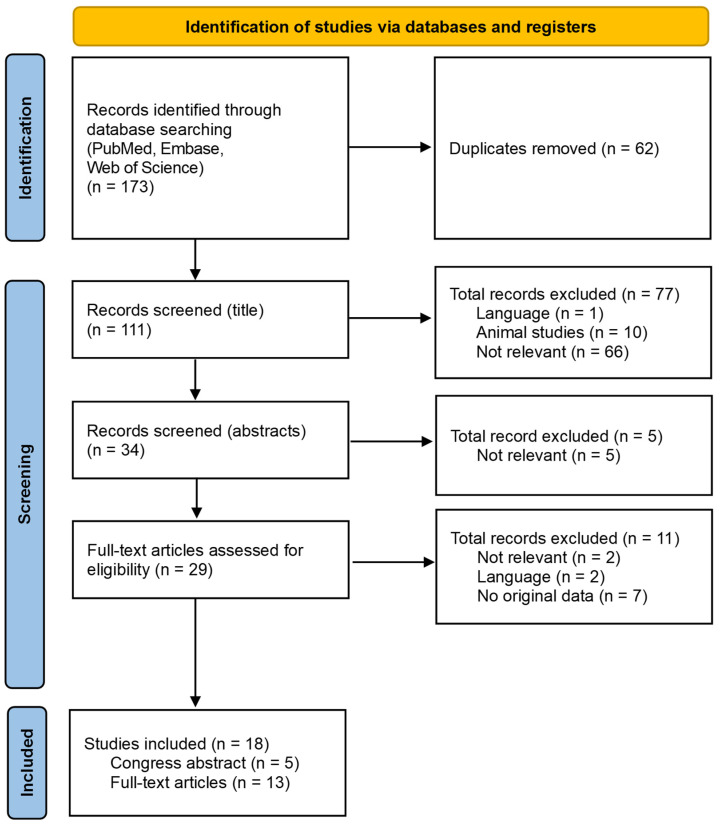
Flowchart of database search and selection of studies.

**Figure 3 ijms-25-03770-f003:**
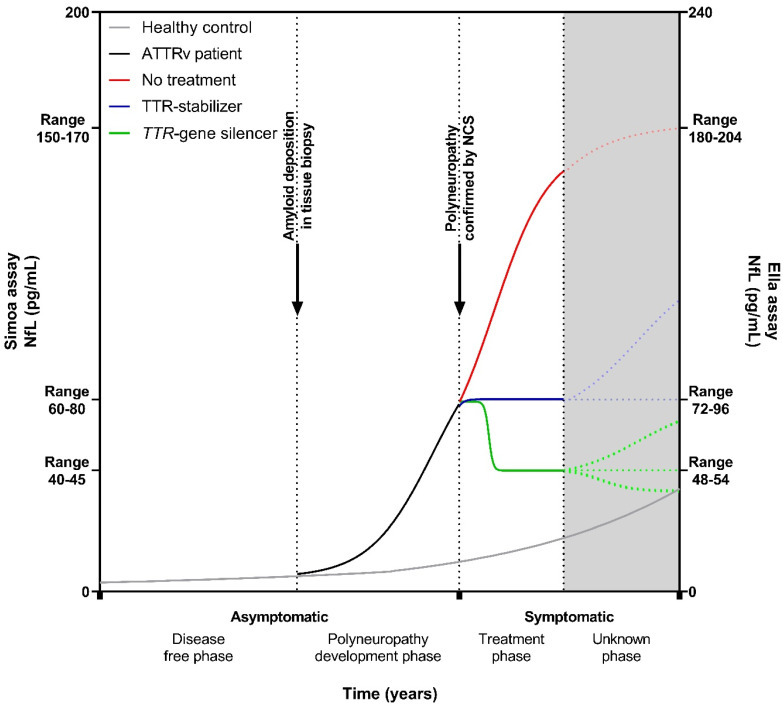
Hypothetical course of neurofilament light chain in a *TTR*v carrier who develops ATTRv amyloidosis with polyneuropathy. NfL levels start to rise when the amyloid is deposited, transforming an asymptomatic *TTR*v carrier into an asymptomatic ATTRv patient. At some point, polyneuropathy can be confirmed by nerve-conduction studies, and the ATTRv patient experiences symptoms. The treatment approach adopted can influence the direction of NfL levels, leading to elevation, stabilization, or reduction. Given the novelty of NfL as a biomarker in systemic amyloidosis and the evolving landscape of treatment modalities, uncertainties persist regarding the long-term course of NfL levels. Gray solid line: healthy control/asymptomatic *TTR*v carrier; black solid line: asymptomatic ATTRv patient; red solid/dotted line: symptomatic ATTRv patient without treatment; blue solid/dotted line: symptomatic ATTRv patient on a TTR-stabilizer; green solid/dotted line: symptomatic ATTRv patient on a *TTR*-gene silencer. NfL: neurofilament light chain; NCS: nerve-conduction studies; ATTRv: hereditary transthyretin amyloid; Simoa: single-molecule array.

**Table 2 ijms-25-03770-t002:** Proposed neurofilament light chain cutoff levels.

Study (Ref.)	Sample Type	Assay	NfL Cutoff Level (pg/mL)	Disease Stage	Sensitivity (%)	Specificity (%)
Loser et al., 2022 [24]	Serum	Simoa	11.7	Asymptomatic and symptomatic	85.7	100
Maia et al., 2020 [33]	Plasma	Simoa	10.6	PND 0 and PND ≥ I	96.2	93.8
			10.6	PND 0 and PND I	92.3	93.8
			66.9	PND I and PND ≥ II (cohort #1)	61.5	92.3
			75.7	PND I and PND ≥ II (cohort #2)	84.6	80.0
Ticau et al., 2021 [34]	Plasma	Simoa	37	Healthy controls and ATTRv-PNP	84.9	94.4
Romano et al., 2024 [39]	Serum	Ella	37.0	Healthy controls and ATTRv-PNP	81.4	98.0
			37.0	Healthy controls and PND I	63.2	98.0
			37.1	Asymptomatic carriers and symptomatic ATTRv patients	81.4	100
			37.1	Asymptomatic carriers and PND I	63.2	100
			57.70	PND I and PND ≥ II	82.4	73.7
González-Moreno et al., 2024 [40]	Serum	ELISA	93.55	Asymptomatic V30M *TTR*v carriers and ATTRv V30M patients	79	87
			92.6	Healthy controls and ATTRv V30M patients	79	80
Carroll et al., 2024 [41]	Serum	Simoa	52.2	PND ≤ I and PND > II	100	55.5
			64.5	Asymptomatic patients and symptomatic patients or sensorimotor converters	92.0	88.5
			88.9	Asymptomatic patients and symptomatic patients and all converters	62.9	96.2

ATTRv: hereditary transthyretin amyloid; ELISA: enzyme-linked immunosorbent assay; Ella: name of a microfluidic cartridge-based immunoassay platform; NfL: neurofilament light chain; PND: polyneuropathy disability; PNP: polyneuropathy; Simoa: single-molecule array; *TTR*v: transthyretin gene variant; and V30M: TTRVal30Met p.(Val50Met).

**Table 3 ijms-25-03770-t003:** Effect of treatment on neurofilament light chain levels.

Study (Ref.)	Effect of No Treatment on NfL Levels Compared to Baseline	Pre-Defined Assessment Time (Months)	Effect of Treatment on NfL Levels Compared to Baseline	Pre-Defined Assessment Time (Months)
Ticau et al., 2021 [34]	APOLLO placebo: ↑	9, 18	APOLLO patisiran: ↓	9, 18
Loser et al., 2022 [24]	No treatment: trend ↑	12	Tafamidis: trend ↑	12
			Patisiran: trend ↑	12
			Initiation of patisiran: trend ↓	12
Sato et al., 2023 [36]			Tafamidis: ↑	Not specified
			Patisiran: ↓	12, 24
Ticau et al., 2023 [38]	APOLLO placebo: ↑	4, 18	APOLLO patisiran → Global OLE patisiran: ↓	4, 18, 30, 42
			Phase II OLE patisiran → Global OLE patisiran: ↓	24, 36, 48
			APOLLO placebo → Global OLE patisiran: ↓	12, 24
Berends et al., 2024 [42]	No treatment and no neuropathy: ↑	27	Diflunisal/tafamidis: =	25
			Patisiran: ↓	14
Conçeicao et al., 2023 [45]			Eplontersen: trend ↓	20
Luigetti et al., 2023 [46]	APOLLO placebo: ↑	4, 18	APOLLO patisiran: ↓	4, 18
			HELIOS-A patisiran: ↓	4, 18
			HELIOS-A vutrisiran: ↓	4, 18
Gilling et al., 2023 [47]	APOLLO placebo: ↑	18	APOLLO patisiran: ↓	9, 18
			APOLLO patisiran → Global OLE patisiran: ↓	9, 54
			APOLLO placebo → Global OLE patisiran: ↓	12, 24, 36
			Phase II OLE patisiran → Global OLE patisiran: ↓	24, 60
Carroll et al., 2024 [41]			*TTR*-gene silencer: ↑	Not specified
			*TTR*-gene silencer: ↓	6

↑: increase; ↓: decrease; =: stable; NfL: neurofilament light chain.

## Data Availability

No new data were generated or analyzed in support of this research.

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
