# Peer review of "Neurofilament Light Chains in Systemic Amyloidosis: A Systematic Review"

_ijms, 2024, doi:10.3390/ijms25073770_

Round 1

Reviewer 1 Report

Comments and Suggestions for Authors

Berends et al present a literature review to assess whether Neurofilament light chain (NfL, a neuron-specific biomarker, that is released into the blood and CSF after neuronal damage) can be used as an early biomarker in patients with systemic amyloidosis and transthyretin gene variant (TTRv) carriers. Whilst this study is small (twelve full-text articles and six abstracts focussing on 36 patients with systemic amyloidosis), the authors conclude that NfL can be used for the early detection of disease onset (marked by rising NfL levels) detecting neuropathy, monitoring treatment effect, and monitoring disease progression in those with systemic amyloidosis. 

Whilst this review highlights the importance of NfL as an early biomarker for disease, the number of studies is small and therefore overinterpretation of the findings should be avoided. Before being considered for publication in this journal, there are a number of points that need to be addressed. These are detailed below.

Abstract / Introduction

·       In the abstract it should be made clear why there is a need for this review. Why now? More emphasis into why this review is important is needed in the abstract. Is there controversy in the field? Are early biomarkers lacking? Why is this important to address. This needs explaining in the abstract and the introduction as this is not clear.  

·       Lines 43-45- a brief overview of the condition is needed- more background about the condition (how it affects people, treatments, cure?) as this emphasises the importance for researching this condition and the importance of publishing such a review. 

·       You mention ‘amyloidosis’ and ‘amyloid’ several times in the introduction without defining it, a definition / description needs to be included for the non-specialist reader eg lines 45-50? Or after line 66.  

·       Lines 73-80 – over what time intervals are you referring to? Be specific e.g line 78, can you add numbers to this?

·       Lines 101-102 – “Lastly, we evaluate if NfL has potential to be implemented in clinical practice in the near future” this is unclear, what do you mean implemented in clinical practice? What specifically needs to be implemented? This needs explaining better.

Methods 

·       Lines 105-106- Where were these clinical studies conducted? Worldwide? This is important due to different regulatory practices undertaken in different countries. Also what years do the studies refer to? Did you search all years? All this criteria needs to be included.

·       Lines 108 and 113 – why exclude patients with Alzheimer’s disease? This needs explaining (ethical reasons or are there links to amyloid beta / Tau pathology?) why are they excluded? Alzheimer’s disease is not mentioned anywhere else in the paper. 

·       Line 142- are they age-matched and sex-matched healthy controls? How old are the subjects involved and is it known whether they have any other related conditions? Line 252 clearly states that NfL levels are influenced by age so this information needs to be included. 

Results / Discussion

·       Table 1 – how are the levels of NfLs being measured? What methods are being used? The same methods need to be used or you cannot compare studies. This information should be included in the legend underneath Table 1. 

·       Figure 2. How were light chain levels measured? a-g outlining what treatment patients were given could be better displayed under figure 2.  

·       Lines 74-76 – how does Ella or Simoa assays differ from ELISA? You mention in line 302 that they are antibody-based assays, but how does the epitope differ and could this account for the differences in NfL levels? Explain. 

·       You mention asymptomatic cases (eg line 57) but what are the symptoms these individuals have? This is important and should be discussed in the introduction and discussion. 

·       Line 123- treatment for what? You mention NfL levels but what was the treatment specifically given for- what condition and / or symptoms?

·       Lines 325-326 – what do you mean ‘burdensome’? Be more specific- are the procedures invasive or cause discomfort to the patients? This needs more explanation. 

·       Lines 337-345- Over what time scale? 

·       Lines 378-382- so how can you reach any conclusions if the studies are not comparable? 

Comments on the Quality of English Language

Some minor typological errors only. Overall English is fine. 

Reviewer 2 Report

Comments and Suggestions for Authors

The authors present a comprehensive and up-to-date analysis of peripheral blood NfL levels as a biomarker of systemic amyloidosis. This addresses a knowledge gap in the field where several studies existed with potentially conflicting results. For this analysis, the studies were selected based on specific selection criteria, and due to the heterogeneity of available data, the authors justifiably stopped short of performing a meta-analysis. Overall the analysis reveal that circulating NfL levels are valuable in prognosticating the disease and assessing treatment response, while being nonspecific and having limited role in assessing disease severity.

Minor comments:

1. The date of database search is stated as "14 February 2023" in the methods, whereas it is mentioned as "14-02-2024" in the abstract. The references contain articles published in 2024.

2. In Fig, 2, subfigures A and B are not labeled.

Reviewer 3 Report

Comments and Suggestions for Authors

In this review, the authors present a detailed analysis of using the Nfl levels as early diagnostic marker in patients with polyneuropathy in comparison with the healthy controls based on the data available from previous literature. Authors clearly discuss the confounding variables and potential drawbacks along with the conclusions giving a full picture of Nfl levles.

Comments:

1. How was systemic amyloidosis confirmed in the study can be included in Table 1 - Clinical examination, electrodiagnostic, quantitative sensory listing?

2. As the Nfl levels vary with age, how did age influence the Nfl levels in the studies included? Were children included in the studies?

3. Is there data available on other secondary central nervous system disorders in the included cases, that could affect the NfL levels? Did the affect the inclusion or exclusion criteria?

4. A note on the various treatment options available in the introduction would be very useful.

5. Note on several assays- simoa, elisa, ella in the introduction would make it more clear.

6. In Table 2, subdivision according to treatment would be good.

7. What's the average treatment time for the drugs to show Nfl changes? Table 3 could include this data if relevant.

8. A discussion on the kidney transplantation affecting the Nfl levels would be good.

Comments on the Quality of English Language

The paper is written in detail and generally in an easy-to-read manner. 

Round 2

Reviewer 1 Report

Comments and Suggestions for Authors

Thank you to the authors for addressing all of my points and queries. The manuscript has now been updated accordingly and I am now happy to recommend this manuscript for publication in this journal.